# QGRAPH-BOUNDED Q-LEARNING:
# STABILIZING MODEL-FREE OFF-POLICY DEEP RL

## ABSTRACT

In state of the art model-free off-policy deep reinforcement learning (RL), a replay memory is used to store past experience and derive all network updates. Even if both state and action spaces are continuous, the replay memory only holds a finite number of transitions. We represent these transitions in a *data graph* and link its structure to soft divergence. By selecting a subgraph with a favorable structure, we construct a simple Markov Decision Process (MDP) for which exact Q-values can be computed efficiently as more data comes in – resulting in a QGRAPH. We show that the Q-value for each transition in the simplified MDP is a lower bound of the Q-value for the same transition in the original continuous Q-learning problem. By using these lower bounds in TD learning, our method is less prone to soft divergence and exhibits increased sample efficiency while being more robust to hyperparameters. QGRAPHs also retain information from transitions that have already been overwritten in the replay memory, which can decrease the algorithm's sensitivity to the replay memory capacity.

## 1 INTRODUCTION

With the wide-spread success of neural networks, also deep reinforcement learning (RL) has enabled rapid improvements in many domains including computer games (Silver et al., 2017) and simulated continuous control tasks (Mnih et al., 2016). Particularly in areas where correct environment models are hard to obtain, such as robotic manipulation, model-free approaches have the potential to outperform model-based solutions (Fazeli et al., 2017; Levine et al., 2016) – as long as enough training data is available or can be generated.

Although efforts in the research community on building an understanding of deep RL for simple examples seem to be rising (Mania et al., 2018), deep reinforcement learning remains underinvestigated from a theoretical point of view. Many algorithms use function approximation, off-policy learning and bootstrapping together – which has even been called *deadly triad* by Sutton & Barto (2018) because this is a very unstable combination of techniques. Although Q-learning is known to have convergence issues even with linear function approximation (Baird, 1995), deep Q-learning descendants like DQN and DDPG often excel empirically (Van Hasselt et al., 2018). At the same time their performance can be unreliable and hard to reproduce (Henderson et al., 2018).

The contribution in this work is two-fold: To add to the community's understanding of when deep Q-learning diverges, we first propose a graph-perspective on the replay memory which allows to analyze its structure and show on educational examples that specific types of structures are linked to divergence. Second, we derive our method from these insights and show that it prevents many cases of soft divergence. Further analyses reveal that this increases sample efficiency, robustness to hyperparameters and preserves information from transitions that have already been overwritten in the replay memory.

## 2 PRELIMINARIES

We consider a standard reinforcement learning setup where an agent interacts in discrete time steps $t = 1, \dots, T$ with an environment that is modeled as a Markov decision process (MDP) with state space $\mathcal{S}$, action space $\mathcal{A}$, initial state distribution $p(s_1)$, transition dynamics $p(s_{t+1}|s_t, a_t)$ and a reward function $r(s_t, a_t)$. In the following, we will assume deterministic transition dynamics.

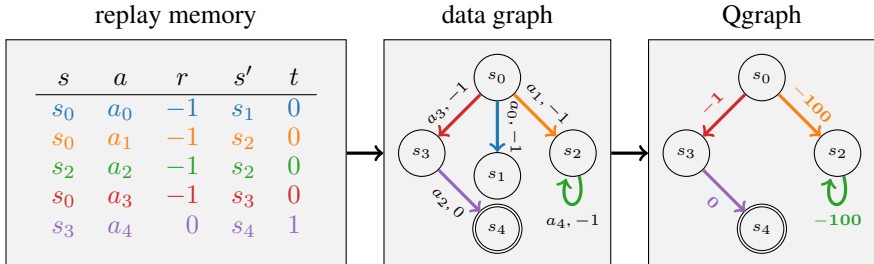

Figure 1: We take a graph perspective on past experience (middle) and extract a subgraph (right) such that its structure allows to compute exact Q-values using Q-iteration for the resulting finite MDP. Its Q-values represent lower bounds to the Q-values in the actual continuous MDP.

At each time $t$, the agent can observe its state $s_t$ and take an action $a_t$ which determines the next state $s_{t+1}$ and an associated reward $r_t$. A policy is a function $\pi$ that maps from states to actions. The sum over future expected rewards when following policy $\pi$ is called return: $R^\pi = \sum_{i=t}^{T} \gamma^{i-t} r_i$, where $\gamma$ is the so-called discount factor. For $\gamma < 1$ and a constant reward $r$ for infinite trajectories, the return forms a geometric series and converges to $\frac{r}{1-\gamma}$. This also means, that if the reward function is bounded by $R_{\min}$ and $R_{\max}$, the smallest and largest possible Q-value can be computed as

$$\left[ \min\left( R_{\min}, \frac{R_{\min}}{1-\gamma} \right), \max\left( R_{\max}, \frac{R_{\max}}{1-\gamma} \right) \right] \tag{1}$$

respectively (Lee & Kim, 2015). The min/max operations are required for terminal states.

Analogously, if the reward only depends on the current state and the agent stays in a state $s$ forever, because action $a = \pi(s)$ does not lead to a change in states, then $\mathcal{Q}(s, a) = \frac{r}{1-\gamma}$. This transfers to larger loops, e.g. if transitions $(s_1, a_1, r_1, s_2)$ to $(s_n, a_n, r_n, s_1)$ are known to be induced by a policy $\pi$,

$$Q(s_1, \pi(s_1)) = \underbrace{r_1 + \gamma r_2 + ... + \gamma^{n-1} r_n}_{r_L} + \gamma^n r_1 + \gamma^{n+1} r_2 + ... = r_L \sum_{t}^{\infty} (\gamma^n)^t = \frac{r_L}{1-\gamma^n} \tag{2}$$

The expected future return for executing an arbitrary action $a_t$ and then following the policy is called $Q^\pi(s_t, a_t) = \mathbb{E}\left[ r_t + \gamma \cdot R^\pi_{t+1} \right]$. The agent's goal is to find the optimal policy $\pi^*$ such that the expected future return is maximized from any state. This can be achieved by finding (a good approximation to) the Q-function and then choosing the action with highest Q-value in each state.

One popular way to learn a Q-function is temporal difference (TD) learning. Given a transition $(s, a, r, s')$, the next Q-value $Q'(s, a)$ is computed based on the current estimate for state $s'$,

$$\mathcal{Q}'(s, a) = r + \begin{cases} 0, & \text{if } s' \text{ is terminal} \\ \gamma \cdot \mathcal{Q}(s', \pi(s')), & \text{else} \end{cases} \tag{3}$$

Such a process with updates that are based on the current estimates of the function to be approximated is called *bootstrapping*. Note that bootstrapping is actually only applied in the case of non-terminal states (i.e. in the second line of the equation). We refer to states that do not require bootstrapping to estimate a Q-value as *anchors*.

In small settings with finitely many states and actions, Q can be represented as a table – this form of Q-learning is referred to as *tabular*. The policy in tabular Q-learning can be read of the table as $\pi(s) = \arg\max_a \mathcal{Q}(s, a)$. In continuous state or action spaces, Eq. (3) can be used with function approximation.

## 3  RELATED WORK

One of the most popular function approximators for Q-functions are neural networks: In deep Q networks (DQN), a single network is trained to take states as an input and predict one Q-value

for each possible action (Mnih et al., 2015). For continuous actions, an actor-critic architecture called deep deterministic policy gradient (DDPG, Lillicrap et al. (2015)) can be used: The critic is represented by one network that computes the Q-value for a given state-action pair. These Q-estimates are used as a training signal for the actor, which is a neural network that represents the policy.

Both DDPG and DQN are model-free algorithms, i.e. do not assume nor learn a model of the environment (including dynamics and other objects in the environment). Furthermore, they use off-policy data, i.e. they store past experience in a replay memory and update their networks based on this experience, even if the policy $\pi$ has changed since the data was collected. It is insightful to note that this replay memory only contains a finite number of transitions $(s, a, r, s', t)$ that all networks are updated from, even for continuous state-action spaces.

The original reasoning behind replay memories and experience replay was to break dependencies between transitions (Mnih et al., 2015), which is important for most function approximation schemes. We will therefore keep the principle of random selection of data for our learning process, but at the same time we will make use of additional information that a graph perspective can provide and would be lost otherwise. Prior work has incorporated trajectory-centric perspectives already: Monte Carlo updates for example do not estimate Q-values based on single transitions as in Eq. (3) but on empirical returns for full episodes. The resulting gradients on the Q-function often suffer from large variance though (Doerr et al., 2019) and therefore many intermediate algorithms exist, mixing TD-learning and Monte Carlo approaches, (e.g. Amiranashvili et al. (2018); Munos et al. (2016)). Trajectories have also been combined to graph structures, for instance to guide exploration (Shkolnik & Tedrake, 2009), to allow for planning (Farquhar et al., 2018) or to re-visit previously discovered states (Dong et al., 2019). Simultaneously to this work, Anonymous (2020) have suggested to build a graph structure from states in the replay memory and thus to propagate information from one trajectory to another if they share a state. Our method also uses cross-trajectory information as a side effect because of the graph perspective it takes on data in the replay memory.

Despite its success in many applications, model-free off-policy deep reinforcement learning can be rather unstable: On the one hand, many algorithms are very sensitive to hyperparameters and even subtle differences in their implementation (Henderson et al., 2018), making it hard to provide sensible empirical comparisons. On the other hand, the theoretical underpinning is quite weak: Reinforcement learning with (even linear) function approximation has already been known to be instable for more than 20 years (Baird, 1995). Since DQN and DDPG combine (highly non-linear) function approximation with bootstrapping and off-policy learning, these algorithms are in a category of methods that Sutton & Barto (2018) call *deadly triad* because it is even more prone to divergence. Removing one of the three critical properties, e.g. by reducing off-policy exploration can therefore have positive effects on learning stability (Kumar et al., 2019; Fujimoto et al., 2019). Our work in contrast focuses on how to stabilize learning with all three properties intact. Other types of off-policy reinforcement learning are much better understood and proven to converge under certain conditions, e.g. for prediction in finite MDPs with non-linear function approximation as in (Maei et al., 2009) or control under the assumption of a stationary policy (Maei et al., 2010). Also simplified MDPs have been examined for reinforcement learning, for instance to combine model-based and model-free methods as in Seijen et al. (2011). We will propose a new type of simplified MDP that is based on novel insights into the crucial role of the data graph structure.

Since empirical success seems to be on the side of DQN and DDPG though, there has been an increasing number of works on simpler reinforcement learning on the one hand (Mania et al., 2018) and analyses of small examples to build an understanding of why this often works so well. Van Hasselt et al. (2018) for instance show that *unbounded* divergence, which would cause floating point NaNs, as expected for the deadly triad, barely happens in deep Q-learning. Instead, typically only *soft* divergence, causing Q-values beyond the realizable range, is observed. Nonetheless, all variants of divergence not just lead to instable Q-estimates but also prolong the learning phase – which is particularly painful for areas with high costs associated to new data samples such as robotics. In this work, we show that many cases of soft divergence are caused by specific data graph structures in TD learning. Our method uses Q-values from the simplified MDP with a favorable graph structure to construct lower bounds for Q-values in the original learning problem.

full example:

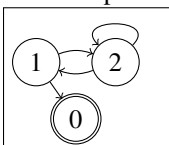

exemplary transition subsets:

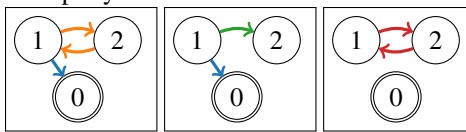

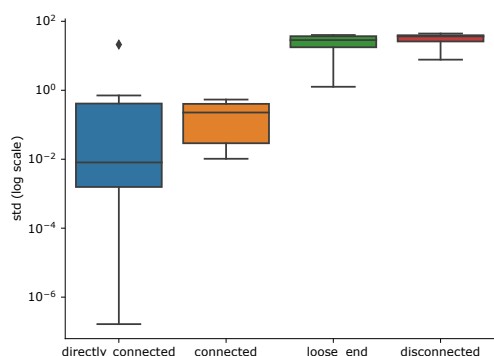

Figure 2: Educational example with four transitions and three states (state 0 is terminal). We characterize transitions based on the graph structure: (in-)directly connected to a terminal state (blue, orange); loose ends (green) and disconnected but infinite paths (red). The right plot illustrates the standard deviation over predicted Q-values for each type of transition.

# 4 EDUCATIONAL EXAMPLES

Let's assume a continuous state-action space, but the replay memory only contains up to four transitions and three states (one of which is terminal). Figure 2 illustrates this as a graph, where each node is a state and each edge represents a transition. Under the assumption that it is possible for an agent to have discovered any subset of transitions, $2^4 = 16$ cases emerge from this setting (one of which is empty and therefore ignored).

We have trained a DDPG critic network for each of the 15 cases under the following conditions: The reward for each step that leads to a non-terminal state is $-1$ (to encourage fast solutions), the reward for reaching the terminal state is 0. The states were assigned 2D-coordinates as from the graph illustration (0: $[0, 0]$, 1: $[-1, 1]$, 2: $[1, 1]$) and the action to move from state $s$ to state $s'$ was defined as the offset $a = s' - s$. For the critic network we chose two layers with 4 hidden states, ReLU activations (except on the output) and Xavier-initialization. The policy was computed as in tabular Q-learning by selecting the action with highest Q-value for a given state. We performed ten thousand updates for each case consisting of all available transitions. The whole procedure was repeated with 10 random seeds that were drawn uniformly from $[0, 1000]$.

Confirming the finding in Van Hasselt et al. (2018), no *unbounded* divergence occurred (which would cause floating point NaNs). However, we found occurrences of *soft* divergence, i.e. Q-values beyond the realizable range as given in Eq. (1).
For further analysis we use the standard deviation of Q-values that were estimated over different random seeds as a measure of soft divergence: if Q-learning for a transition converges, all Q-values should be identical and thus have a standard deviation close to zero. The more soft divergence occurs however, the larger the standard deviation becomes. Even if all trials diverge, it is highly unlikely that the resulting Q-values are identical.
Evaluating the distribution of standard variations reveals a link between the structure of the Q-graph and soft divergence:
Transitions $(s, a, r, s', t)$ where $s'$ is terminal are referred to as *directly connected*. Their Q-values are estimated almost perfectly, because Q-learning is reduced to supervised learning in these cases (cf Eq. (3)).
Transitions that end in a non-terminal state from which a terminal state is reachable are referred to as *connected*. Their Q-estimates exhibit only slightly more variance than the directly connected transitions. Presumably the reachable terminal state still acts as an anchor for the Q-value (as long as all transitions on the path are regularly used for updates). In line with this hypothesis, the two following categories that do not have an anchor show significantly more variance in their predictions:

If no terminal state is reachable from $s'$ and there is no infinite path from $s'$, the transition is referred to as a *loose end*. These transitions occur for instance at the end of each episode in episodic learning setups, when the agent is reset to a starting position. It is insightful to note that Q-values for such transitions are conceptually ill-defined in tabular Q-learning where a state without successors would be defined as terminal. For non-terminal states, a Q-value could be determined under the assumption that further transitions exist (and just have not been experienced yet), but then the Q-value is

estimated using bootstrapping from another Q-value that has never been explicitly updated. This may be okay, if the learned function generalizes nicely but there is no guarantee for this to happen. Transitions are referred to as *disconnected* if no terminal state is reachable from $s'$ but there exists at least one infinite path from $s'$. These transitions caused the highest variance in Q-estimates. In contrast to loose ends however, the Q-value for these transitions is well-defined under the assumption that all possible transitions are known and can even be computed analytically (cf. Eq. (2)). Even in our educational toy example, where the replay memory was not changed during training, the Q-values for disconnected transitions were rarely found and caused soft divergence instead.

In our method, all available exact Q-values (under the assumption that all possible transitions are known), including those for disconnected transitions, will be used as a lower bound for the Q-value in the original learning problem. We will show empirically that, besides further effects, this reduces the variance of predicted Q-values.

## 5 METHOD

Despite the continuous state-action space, the networks in DDPG are updated based on a finite set of transitions from the replay memory. It is therefore possible to take a graph perspective on this data: A transition $(s, a, r, s', t)$ can be seen as an edge between the nodes corresponding to states $s$ and $s'$ (which is terminal iff indicated by $t$); and can be annotated with $r$ and $a$. Any hashing function can be used to encode nodes and detect if the same node is revisited. For small spaces, it is enough to use the state vector directly. This is not supposed to introduce any discretization beyond the limits of precision. We refer to this directed graph as *data graph* (see Figure 1 for an illustration).

### QGRAPH

Building on the insights from section 4, we select the largest set of transitions from the *data graph* for which exact Q-values can be computed under the assumption that the resulting graph is complete (i.e. that all possible transitions and all states are included). That is, we extract all transitions from the data graph except for loose ends. Formally, this induces a smaller finite MDP for which the associated Q-function can be computed using tabular Q-iteration with guaranteed convergence due to its contraction property. Our method is agnostic to the algorithm that computes these Q-values, so for instance it is also possible to solve the linear equation system for a sparse transition matrix. In any case, the computational overhead to compute these Q-values can depend on the number of transitions in the replay memory, but it is independent of the input dimensionality. We annotate the subgraph with the resulting Q-values and refer to it as QGRAPH.

In many settings, there are known *zero actions* $a_z$ that do not change the agent's state at all, e.g. moving by 0 units or applying 0 force. If those are applicable in all states, it may be possible to add a self-loop to every single node in the data graph. This effectively eliminates all lose ends and turns them into disconnected states, in other words it enables the QGRAPH to contain all transitions from the data graph.

### QGRAPH values as lower bounds on the original Q-learning problem

The original MDP contains more states or transitions than the QGRAPH unless the environment has discrete state-action spaces and the replay memory contains at least one sample for each possible transition. Then, the Q-values do not transfer to the original MDP as a correct solution but in deterministic environments, they can serve as a lower bound for the true Q-values:
Each Q-value $\mathcal{Q}(s, a)$ on the QGRAPH is based on actually experienced trajectories, but it is possible that unseen states and transitions exist. Assume that transitions $(s_0, a_1, r_1, s_1)$ and $(s_1, a_2, r_2, s_2)$ are already known and part of the QGRAPH, but in fact at least one further transition from $s_1$ exists. Then the Q-value for the first transition is lower bounded by $\mathcal{Q}(s_1, a_2)$ because of the $\max$ operation:
$\mathcal{Q}(s_0, a_1) = r + \max_a \mathcal{Q}(s_1, a) \geq r + \mathcal{Q}(s_1, a_2)$
Thus, each Q-value for a transition in our QGRAPH is a lower bound of the Q-value for the same transition in the original MDP on continuous state and action spaces.

If the environment is non-deterministic, less tight bounds can be established under additional assumptions. For instance, let's assume that for all states and any given series of actions $\mathfrak{A}$, the empirical returns $R_i$ that an agent can observe when following $\mathfrak{A}$ differ by at most $\delta$. Then all Q-values from the simplified MDP apply as lower bounds with margin $\delta$: $\mathcal{Q}_{\text{true}}(s, a) \geq \mathcal{Q}_{\text{simplified}}(s, a) - \delta$.

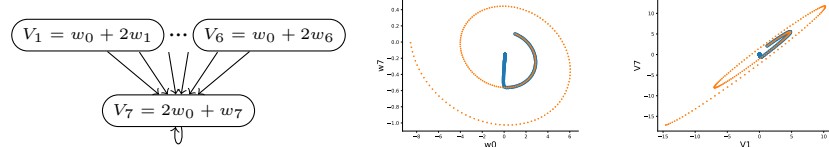

Figure 4: The 7-state star problem (Baird, 1999). Using vanilla TD-learning, state values and weights spiral out to infinity (orange dots). Applying our graph-based lower bounds however makes TD-learning converge to the correct solution (blue solid line).

**QGRAPH-bounded Q-learning**

Q-values from our QGRAPH can be used as lower bounds in bootstrapping for temporal difference learning as in Eq. (3) – a method we call QGRAPH-*bounded Q-learning*:

$$\mathcal{Q}'(s, a) = \max(\text{LB}, r + \gamma \cdot \max_{a'} \mathcal{Q}(s', a')) \tag{4}$$

where LB represents a per-sample lower bound. Loose ends are not associated with such a lower bound but could potentially still carry important information (e.g. in cases where the function approximator has generalized nicely to the states involved). The transitions associated with loose ends can therefore still be used as usual, i.e. using unbounded TD-learning.

## 6 EXPERIMENTAL RESULTS

We evaluated the core of our method on a classical toy example for convergence issues in value learning (Section 6.1). The simulated continuous control problem for all further experiments is described in Section 6.2. We first present an overview of learning performance with a focus on sample efficiency and robustness to hyperparameters in Section 6.3. In Section 6.4, we verify that the observed effects are in line with the guiding hypothesis from our educational example in Section 4. Finally, we examine additional aspects of our method: the impact of zero actions and more trivial bounds on Q-values (Section 6.5) as well as the method's interaction with limited replay memory capacity (Section 6.6).

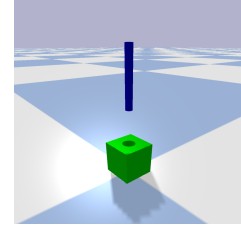

Figure 3: Simulated Peg-In-Hole task.

### 6.1 BAIRD'S STAR PROBLEM

The 7-state star problem (Figure 4) was proposed by Baird (1999) to demonstrate convergence issues in value iteration with (linear) function approximation. The agent receives a reward of zero for each action and thus the correct solution to the problem is to set all weights to zero and obtain state-values of zero. If all weights are initially positive and $w_0$ larger than the others, this causes oscillatory behavior of both state values and weights. We reproduced the exact setting and result plots for Figure 4.2 in Baird (1999). Applying our graph view to the problem, we can derive a lower bound of zero for $V_7$ because it has a self-loop with reward 0; and thus this lower bound recursively leads to a lower bound of $0 + \gamma V_7 = 0$ for all other states. These graph-based bounds can be applied in TD learning in analogy to Eq. (4) as $V'(s) = \max(LB, r + \gamma V(s'))$. As a result, our method converges to the correct state values rather than diverging to infinity as Figure 4 illustrates.

### 6.2 EXPERIMENTAL SETUP

All further experiments were conducted on a simulated continuous control task, see environment details in Appendix A.1 and DDPG implementation details in Appendix A.2. After each epoch, the Q-targets were updated, i.e. no explicit target network was used, since those are known to prolong training and thereby postpone convergence issues but not solve them (Van Hasselt et al., 2018). All algorithms were tested for 300 episodes. We tested vanilla DDPG on a grid of learning rates for actor and critic in $\{10^{-2}, 10^{-3}, 10^{-4}\}$ and chose three cases that are representative for the spectrum of DDPG performance. The learning curves for all learning rates are shown in Appendix 7. Most results will be presented in the form of learning curves, where the line represents the mean performance over ten runs with different random seeds and the shaded area highlights the standard deviation of the mean estimator, i.e. $\frac{\sigma}{\sqrt{n}}$.

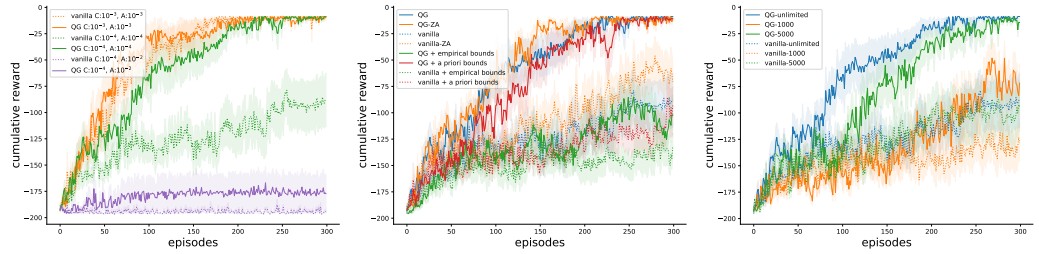

Figure 5: Learning curves for vanilla DDPG and QGRAPH-bounded Q-learning('QG') on the left; a number of baselines in the center; and performance under limited graph capacity on the right.

### 6.3 SAMPLE EFFICIENCY AND ROBUSTNESS TO HYPERPARAMETERS

We hypothesized that QGRAPH-based lower bounds would correctly limit the range of Q-values which prevents some cases of soft divergence and thereby increases sample efficiency. We further hypothesized that explicit bounds would barely have any impact in cases when vanilla Q-learning works well, because our method as described in Eq. (4) reduces to standard TD learning when no bound is violated. In other words this implies that QGRAPH-bounded Q-learning should never decrease performance.

For a first overview, we compared learning curves of QGRAPH-bounded Q-learning ('QG') to those of vanilla DDPG in Figure 5 (left). As expected, QGRAPHs speed up learning for all examined learning rates. The effect size varies and is larger for those learning rates that lead to relatively poor performance in vanilla DDPG. This decreases the gap in performance between different learning rates and can therefore be interpreted as an indicator for increased robustness to hyperparameters.

### 6.4 VARIANCE OF PREDICTIONS

To assess if this performance is due to similar effects as in our educational examples, we evaluated the variance in predicted Q-values at the end of each experiment under the learning rate with largest effect size ($10^{-4}$). We covered the state space with a regular grid of 27 states and evaluated the learned Q-value for each of these states with a set of eleven given actions as well as with the action that the actor network suggests for each state. The full list of all states and given actions that were tested can be found in Appendix A.4.

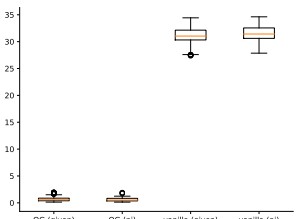

Figure 6: Standard deviation of predicted Q-values.

For the boxplot in Figure 6, we collected the standard deviations over the predicted Q-values for each state-action pair from 10 runs with different random seeds. The orange line indicates the median value, the box extends from the lower to the upper quartile value, the whiskers cover 1.5 times the inter quartile range and outliers are shown as circles. The results shows very clearly that QGRAPH-runs resulted in significantly less variance for predicted Q-values, indicating that QGRAPH-bounded Q-learning does indeed prevent cases of soft divergence.

### 6.5 FURTHER BASELINES

We ran the following baselines to deepen our understanding of the previously reported effects:

In many settings a *zero action* is known that does not change the agent's state (in our case it is the offset in position by zero meters). Adding hypothetical transitions with the zero action after each physical transition ('vanilla-ZA') improves the structure of the data graph by turning loose ends into disconnected transitions. Using zero actions in our method ('QG-ZA') not only improves the structure of the data graph but also spreads information in the form of lower bounds to predecessors in the QGRAPH. The results in the center of Figure 5 show that zero actions improve marginally over vanilla DDPG and QGRAPH-bounded Q-learning. This implicates on the one hand that the structure of the data graph matters and that in particular vanilla DDPG benefits from a structure without loose ends. At the same time, adding our QGRAPH is much more effective than changing

the data graph only. This demonstrates that the propagation of information through the QGRAPH and the integration of lower bounds into TD-learning are the main contribution of our method.

The next set of baselines was designed to evaluate how much influence the exact bounds have. Bounded temporal difference learning could, besides our QGRAPH-based bounds, integrate two further types of lower and upper bounds: *A priori* bounds may be known in the case of a bounded reward function, see Eq. (1). *Empirical* bounds may seem like an alternative for correct a priori bounds: rather than using known bounds on the reward, these bounds could be estimated from experience. Note that the true Q-values are guaranteed to lie within QGRAPH-based bounds and correct a priori bounds, while empirical bounds might be too tight. We combined QGRAPH-bounded Q-learning and vanilla DDPG with both types of bounds. When several bounds were available for one Q-value, the tightest upper and lower bound were chosen. The results in Figure 5 confirm that incorrect empirical bounds (green lines) have adverse effects on both methods, while a priori bounds do not seem to have any significant effect. We conclude that the tight sample-specific lower bounds from our QGRAPH are key and much more informative than more general bounds.

### 6.6 LIMITED GRAPH CAPACITY

In deep reinforcement learning, the replay memory is typically a FIFO-buffer ('first in, first out'), i.e. those elements that were added first are overwritten first when the buffer is full. For a data graph, it is possible to delete single transitions but there are two possible effects: On the one hand, some information from deleted transitions can be implicitly contained in its predecessors' Q-values on the QGRAPH, which could imply that our method is more robust to small memory capacities. On the other hand, cuts from deleted transitions can stop information propagation through the QGRAPH, which could in turn slow down further progress. We therefore empirically compared the drop in performance for vanilla DDPG and our QGRAPH-bounded Q-learning with graph capacities of 1000 and 5000 transitions. For comparison, the average unlimited graph contained roughly 30,000 unique transitions at the end of our 300 episode experiments. As the right plot in Figure 5 illustrates, a QGRAPH-based method that is limited to only 1000 samples still performs on par with unlimited vanilla DDPG, while the vanilla DDPG performance decreases for a limit of 1000 transitions. Higher graph capacities do not seem to impact performance significantly.

## 7 CONCLUSION

From the observation that even for continuous state and action spaces, model-free off-policy deep reinforcement learning algorithms perform network updates on a finite set of transitions, we have developed a graph perspective on the replay memory that allows closer analysis and induces a new MDP under the assumption that the replay memory is complete. In this setting, two types of structures are most likely to lead to soft divergence: non-terminal states without successors, for which the Q-value is conceptually ill-defined; and infinite loops with no path to a terminal state.

It is possible however to construct a simplified MDP from a subgraph for which all Q-values are well-defined and can be analytically computed. In deterministic environments, these Q-values are lower bounds for the Q-values of the original MDP; in probabilistic environments some further assumptions are required to apply lower bounds. Enforcing these bounds in TD-learning solves Baird's classical star problem and empirically prevents cases of soft divergence on a continuous control task. The bounds can also be seen as a way to spread information back along the full trajectory without the increase in gradient variance that is linked to Monte Carlo Backups. The largest effect was found for those hyperparameters that lead to worst performance, supporting the view that our method prevents some degenerate cases in function approximation for model-free off-policy deep reinforcement learning. Additionally, the QGRAPH that holds the lower bounds, provides an additional kind of memory for information from transitions which have already been overwritten in the replay memory.

This work gives rise to a series of questions for future work: there may be tighter bounds including data-driven upper bounds, the formulation for probabilistic environments may be extended to cover further cases, and the reward function most likely interacts with soft divergence and indirectly with our method. We may thus examine potential implications for reward shaping in the future.

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

# A    APPENDIX

## A.1    PEG-IN-HOLE ENVIRONMENT DETAILS

The environment was implemented using pybullet[1] A blue peg is supposed to be inserted into a green object (see Figure 3). The peg is always upright and velocity-controlled: an action represents the three-dimensional offset to the next position. The simulation is stepped forward until a stable new position is reached. The actions are box-constrained to $[-1, 1]$ in each dimension which corresponds to a movement of 1cm. The green object has a width of 5cm and is within a cubic state space of width 20cm. The peg has a diameter of 1cm, the hole's diameter is 2cm. The agent receives a distance-based reward $r = \exp(-\frac{\delta}{0.03}) - 1$, where $\delta$ is the Euclidean distance to the goal position in meters.

## A.2    NETWORK DETAILS

The critic network consists of three fully connected layers with 200 nodes each. For the inner layers, ReLU activations were used. The network was initialized with weights sampled from a normal distribution of mean 0 and std 0.001. The actor network also consists of three fully connected layers with 200 nodes each, but used tanh activations and was initialized from a He-uniform distributions (He et al., 2015).

All neural networks were implemented using tensorflow[2] and optimized using the AdamOptimizer, with 50 training epochs after each episode (i.e. 200 agent steps) and up to 15 random mini batches of data per epoch.

## A.3    DETAILED PERFORMANCE FOR DIFFERENT LEARNING RATES

We ran vanilla DDPG on a grid of learning rates where both actor and critic learning rates is chosen from $\{10^{-2}, 10^{-3}, 10^{-4}\}$. We chose the three curves with solid lines as representative for the spectrum of performance and based all further evaluation on these learning rates.

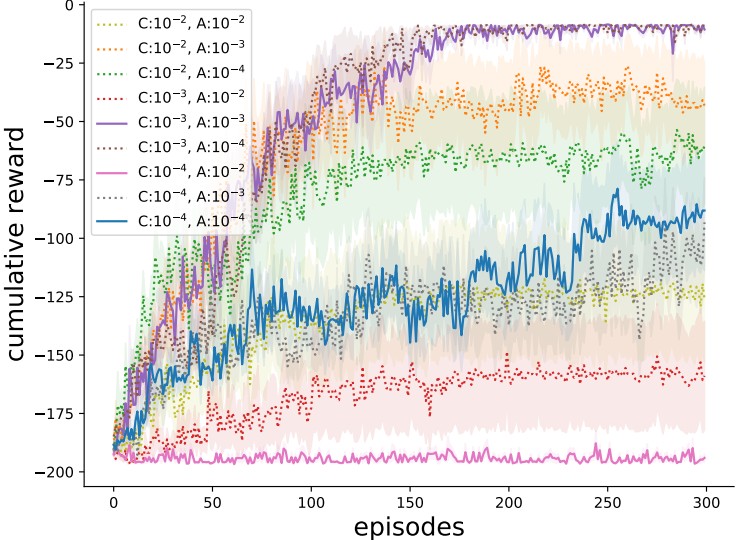

Figure 7: Vanilla DDPG performance for all tested learning rates.

[1] https://github.com/bulletphysics/bullet3
[2] www.tensorflow.org

### A.4 STATE-ACTION PAIRS TESTED FOR VARIANCE IN PREDICTED Q-VALUES

Here we show all states and actions that were used to examine the variance in Q-estimates in Figure 6.

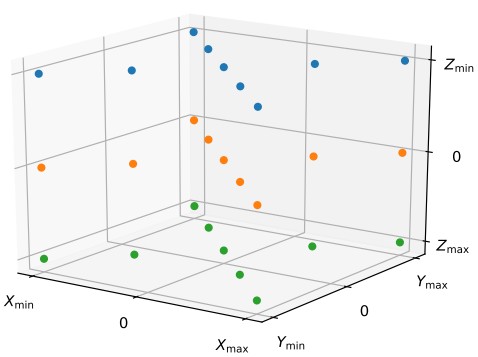

| $a_x$ | $a_y$ | $a_z$ |
|------|------|------|
| 0 | 0 | 0 |
| 0 | 0 | 1 |
| 0 | 0 | -1 |
| 0 | 1 | 0 |
| 0 | -1 | 0 |
| 1 | 0 | 0 |
| -1 | 0 | 0 |
| 0.5 | 0.5 | 0.5 |
| -0.5 | -0.5 | -0.5 |
| 1 | 1 | 1 |
| -1 | -1 | -1 |

