# OpenReview forum: "Qgraph-bounded Q-learning: Stabilizing Model-Free Off-Policy Deep Reinforcement Learning"
_ICLR.cc/2020/Conference — Reject_

### Official Review · AnonReviewer1 · 2019-10-22
**Official Blind Review #1**

**Rating:** 3

**Review:**


The paper proposes Qgraph, an algorithm that  addresses the problem of extrapolation error that appear in RL tasks with continuous action spaces. The authors describe a method to construct a graph from transitions generated by some policy. In this graph nodes correspond to states and (s, a, r, s’, t) define transitions between these nodes. Then this representation is simplified and  used to compute Q-values using methods for tabular MDPs.

The related work section is missing several methods that attempt to address the same problem. Batch Constrained Q-learning (Fujimoto etal, 2018) introduces a formulation of Q-learning that constrains action selection with a generative model trained on a replay buffer in order to omit unseen actions. BEAR-QL (Kumar etal, 2019) describes a similar approach that uses a hard constraint based on MMD. It would be interesting to discuss connections with the recent work on off-policy batch RL.

The clarity of the paper can be improved. In particular, I have several questions regarding the method:
1) How are node of the graph are constructed? Does one state correspond to a single node or several states are merged into a different node?
2) How the actions are selected?
3) What are the assumption regarding the initial state distribution? Does the set of initial states have to be finite?
4) If two similar states appear in different branches of the graphs, are the corresponding nodes merged or not?
5) If the considered environments are deterministic, what is the motivation for stochastic approximation of dynamic programming?

The approach has several major limitations. One of the main limitations of the approach is that it can be applied only to deterministic tasks. Although it is not stated clearly in the paper, it seems also requires to have a finite set of initial states.

The experimental evaluation is performed on a limited set of tasks and it is rather unclear whether the method can be scaled to higher dimensional control problems.

Overall, I feel that the paper needs to be significantly improved.


**Experience Assessment:**

I have published one or two papers in this area.

**Review Assessment: Checking Correctness Of Derivations And Theory:**

I assessed the sensibility of the derivations and theory.

**Review Assessment: Checking Correctness Of Experiments:**

I assessed the sensibility of the experiments.

**Review Assessment: Thoroughness In Paper Reading:**

I read the paper at least twice and used my best judgement in assessing the paper.

---

> ### Author Response · Authors · 2019-11-11
> **Answer to Review #1**
>
> Thank you for your clear review and suggestions/questions. While we are re-writing parts of the paper in the background, we'd like to answer your questions already. Eventually we will integrate these answers into the paper such that it clearly communicates these points from the beginning.
>
> The two pieces of related work you pointed out seem related indeed. We will discuss them in the next version of our paper. These approaches are complimentary to our work in the sense that they also point out aspects of Q-learning that lead to instabilities. However, they focus on the off-policy property and suggest ways to constrain the actions that are selected. In our work, the action selection is not constrained or altered at all. Instead, we introduce bounds to the Q-function that we enforce during learning. It may actually be of interest for future work to investigate how these methods can be combined and whether this further stabilizes Q-learning.
>
> 1) One state does correspond to a single node. We never merge different states into one node, but of course this is a matter of floating point precision. In our implementation, states are measured in meters and rounded to 4 digits.
> 2) Our method focusses on learning a Q-function. How actions are selected is not altered by our method and therefore any existing method may be plugged in. For our experiments, we use the standard DDPG setup in which an actor network is trained and used as a policy. For exploration, we add Gaussian noise to the output of the actor network.
> 3) There is no assumption about the initial state distribution. One of the most important insights of our work is that state-of-the-art DDPG and DQN only work on samples in the replay memory. These samples may originate from a problem with any initial state distribution, but the full distribution is never used - only the finite number of samples that is stored in the replay memory. Note that these model-free methods do not execute any virtual rollouts (since no model is learned that could be used). If you could point us to the part of our paper that made you think there were constraints to the distribution of initial states, we'd love to clarify or re-write that paragraph.
> 4) If they are actually the same state (up to precision), the nodes are merged; enabling a cross-over of experience from differen trajectories.
> 5) The considered environments are deterministic. We are not sure what you refer to as "stochastic approximation"? Dynamic programming is used to obtain Q-values for the simplified MDP - however, any other way to obtain the correct Q-values works with our algorithm.  Function approximation is used because the state-action space is continuous.
>
> The approach is indeed limited to deterministic tasks (although smaller violations to this assumptions may not be practically relevant).
> There are ways to extend our work to non-deterministic tasks that we will also discuss in the paper:
>
> If the environment is non-deterministic, less tight bounds can be established under additional assumptions.
> For instance, let's assume that for all states and any given series of actions \mathfrak{A}, the empirical returns R_i that an agent can observe when following \mathfrak{A} differ by at most \delta.
> Then all Q-values from the simplified MDP apply as lower bounds with margin \delta:
> Q_true(s,a) >= Q_simplified(s,a) - delta
>
>
> Our method works on the graph structure only.
> It is possible to build a similar graph from high-dimensional input, see for instance the graph of image inputs in this ICLR submission we discovered recently: https://openreview.net/pdf?id=HkxjqxBYDB
> Given the graph structure, our method only depends on the number of nodes and edges in the graph but it is independent of the dimensionality of the original input.
> We will add more details on the complexity of our algorithm in the paper.
> Another question is whether a graph structure is still useful in high-dimensional spaces, because it may seem that it is even less likely for the same node to reappear: (1) we believe that in most cases, only a manifold of this high-dimensional space is actually visited (e.g. only some specific type of image), (2) there are typical attractor-states in many cases, e.g. in the corner of the state space or along edges of an obstacle, (3) in off-policy algorithms you can actively use exploration to re-visit states, (4) overall we have the impression that passing information along full trajectories is more useful than the cross-trajectory information exchange. Note that we pass this information without introducing the high gradient variance that is typically linked to full trajectory backups in Monte Carlo methods!
>
> Do these comments fully answer your questions, or are there any follow-up questions? Thanks!

---

> > ### Author Response · Authors · 2019-11-11
> > **One more comment**
> >
> > Adding to the question whether our graphs are also useful if states are never re-visited.
> > This touches on the question if we can have sufficiently many nodes in the qgraph to derive lower bounds. A 'cheat' related to this issue that we introduce in the paper are zero actions: often some action is known to not change the state (e.g. apply zero force). Then, this zero-actions can be applied to any state and introduce self-loops. Effectively this means that loose ends (which cause highest variance in our educational example) are eliminated.

---

### Official Review · AnonReviewer2 · 2019-10-23
**Official Blind Review #2**

**Rating:** 6

**Review:**

This paper is trying to tackle the soft divergence issue in deep RL when algorithms combine function approximation, off-policy learning and bootstrapping, which is also called deadly triad by Sutton & Barto (2018). The paper proposes a way to represent the transitions in the replay memory as a data graph, then construct a simple MDP from it. Much more accurate Q values could be computed from the simple MDP and it provides a lower bound for the Q-values in the original problem. In this way, the method becomes less prone to soft divergence.

The idea of constructing a smaller MDP whose Q-values can be computed exactly by dynamic programming on tabular states, then use these Q-values to help dealing with the instability issues in deep RL is very interesting. In the rebuttal, I'd like the authors to address my major concern of the paper, where the proposed method seems to assume that the finite number of transitions could form a graph, which might not be always true. In typical continuous state spaces, the same state might not appear twice in the sampled transitions. In these cases, the graph becomes a number of disconnected chains and the Q-values from this MDP might not be accurate. Maybe I'm missing something, it's not very clear to me how the proposed method could be applied in the common case in deep RL where there's seldom a loop and the states are rarely visited twice.

**Experience Assessment:**

I have published one or two papers in this area.

**Review Assessment: Checking Correctness Of Derivations And Theory:**

I assessed the sensibility of the derivations and theory.

**Review Assessment: Checking Correctness Of Experiments:**

I assessed the sensibility of the experiments.

**Review Assessment: Thoroughness In Paper Reading:**

I read the paper at least twice and used my best judgement in assessing the paper.

---

> ### Author Response · Authors · 2019-11-11
> **Answer to Review #2**
>
> Thank you for your comments.
> We are now re-writing parts of the paper and will integrate the following answer eventually to make sure the paper already conveys these messages:
>
> You are right in your concern that the graph may consist of several disconnected components (or chains, as you call them).
> Actually, re-visiting states happens slightly more often than you may think: at corners of the state space, at edges of an obstacle, at narrow goal areas. However, of course, this depends on the environment and may not happen at all. One technique we propose in the paper to deal with this are zero-actions: often some action is known to not change the state (e.g. apply zero force). Then, this zero-actions can be applied to any state and introduce self-loops.
> Additionally, in off-policy algorithms you can use exploration to re-visit states. Overall we have the impression that passing information along full trajectories is more useful than the cross-trajectory information exchange at re-visited states. Note that we pass this information without introducing the high gradient variance that is typically linked to full trajectory backups in Monte Carlo methods.
>
> The Q-values we derive from the simplified MDP are only computed on the Qgraph, which is a subgraph of the data graph. Thus, the 'chains' you mentioned would not be included in the Qgraph. We would still use all samples to train the Q-function on the original problem though, just the number of samples for which a lower bound can be provided, varies. If zero actions are used, there is at least one lower bound for every sample.
>
> Does this make sense to you, do you have any further or follow-up questions?

---

### Official Review · AnonReviewer3 · 2019-10-23
**Official Blind Review #3**

**Rating:** 1

**Review:**

The paper aims to build an understanding of deep RL. Because RL remains under-investigated from a theoretical point of view. Many algorithms use function approximation, off-policy learning and bootstrapping together--This is an unstable combination of techniques. In this paper, the authors propose a graph-perspective on the replay memory which allows to analyze the structure of deep RL.

The paper aims at an important issue in deep RL. The motivation of the paper is meaningful.
The paper gives a good summary of the previous related works in Section2.

The paper in the current form needs to be polished again. To obtain a better score, I suggest the authors to modify this paper in these ways: First, the introduction section needs to provide more details, including the pros and cons of previous related works on the research problem of this paper, the challenges you face when dealing with this issue and the contributions; Second, there were more than a few spelling and grammatical errors, please proofread the work and improve the writing; Third, the paper lacks logic in writing. The writing from Section.2 to Section.6 needs to be organized better. It is difficult for readers to grasp the key ideas of the paper through a quick assessment.
The paper focuses on the understanding of RL when deep Q-learning diverges, however, most of the conclusions in the paper are not based on the necessary theoretical proof, but the observations on the experiments.

It would be better if this paper can provide a clear illustration for the proposed method as well as the experiments section.

**Experience Assessment:**

I have read many papers in this area.

**Review Assessment: Checking Correctness Of Derivations And Theory:**

N/A

**Review Assessment: Checking Correctness Of Experiments:**

I did not assess the experiments.

**Review Assessment: Thoroughness In Paper Reading:**

I made a quick assessment of this paper.

---

> ### Author Response · Authors · 2019-11-11
> **Answer to Review #3**
>
> Thank you for those comments.
> As you suggested, we will of course re-iterate the paper to find typos and grammatical errors; add more details on related work, include more explicit pros/cons; explicitly add a list of our contributions (graph-perspective on the replay memory; thereby insights into different classes of nodes and their impact on convergence; the q-graph-based bounds for Q-learning which have a set of positive effects).
> Also the conclusions will be extended to better link back to the theoretical insights from section 4.
>
> For the remaining points you raised, we would like to confirm our understanding of your comments:
> 1. "the challenges you face when dealing with this issue". The challenge we face is soft divergence, but that is a widely known issue. Do you think the paper would benefit from re-iterating more contents from prior work on soft divergence in Q-learning?
> 2. Would pseudo-code for our method, including the replay memory, data graph, q-graph and network training, provide a "clear illustration for the proposed method"?
> 3. What could a similar illustration for the experiments section look like? Do you think an additional paragraph in the beginning of the experiments section that describes the subsection strucuture would be helpful?
>
> If you feel that any of your requests is not reflected in our suggestions here, please let us know. Thanks.

---

### Decision · Program_Chairs · 2019-12-19

**Decision:**

Reject

**Comment:**

This paper proposes a method to reduce the instability issues of off-policy deep reinforcement learning.  The proposed solution constructs a simple MDP from the experience in the agent's replay memory.  This graph is used to compute a lower bound for the values from the original problem. Incorporating this bound can make the learning system less prone to soft divergence.

The reviewers appreciated the motivation of the paper and the direction of this research.  However, the reviewers were not convinced that the formulation was sufficiently complete.  There were concerns that the method makes additional assumptions about the data distribution (the presence of state aggregation and the absence of repeated states in continuous spaces).  Reviewers found related work was missing.  The reviewers also found multiple aspects of the presentation unclear even after the author response.

This paper is not ready for publication as the generality of the proposed method was not sufficiently clear to the reviewers after the author response.